# A 40 MHz 11-Bit ENOB Delta Sigma ADC for Communication and Acquisition Systems

**DOI:** 10.3390/s23010036

**Published:** 2022-12-20

**Authors:** Hussein Fakhoury, Chadi Jabbour, Van-Tam Nguyen

**Affiliations:** 1Scalinx, 75013 Paris, France; 2Telecom Paris, 91120 Palaiseau, France

**Keywords:** analog to digital conversion, Delta Sigma modulators, CMOS design, decimation filter

## Abstract

This paper describes a Delta Sigma ADC IC that embeds a 5th-order Continuous-Time Delta Sigma modulator with 40 MHz signal bandwidth, a low ripple 20 to 80 MS/s variable-rate digital decimation filter, a bandgap voltage reference, and high-speed CML buffers on a single die. The ADC also integrates on-chip calibrations for RC time-constant variation and quantizer offset. The chip was fabricated in a 1P7M 65 nm CMOS process. Clocked at 640 MHz, the Continuous-Time Delta Sigma modulator achieves 11-bit ENOB and 76.5 dBc THD up to 40 MHz of signal bandwidth while consuming 82.3 mW.

## 1. Introduction

Advanced mobile communication standards, such as 5G NR and 6G, require Analog to Digital Converter (ADC) with wide signal bandwidth over several tens of MHz and high dynamic range (DR). Other applications, such as medical imaging, video and instrumentation, also demand such ADCs. Several architectures are suited to implement these specifications with very competitive figures of merit such as Successive Approximation Register (SAR), pipeline and Continuous Time (CT) Delta Sigma (ΔΣ) [1]. SAR and pipelined ADCs were typically used in these scenarios because they offer a good compromise between resolution and speed. Compared to classical Nyquist converters, CT ΔΣ ADCs have been investigated and drastically improved towards high conversion rates [2]. Due to their implicit anti-aliasing filtering (AAF), robustness to interference, and resistive input impedance, they have achieved a dominant position for digitization in wireless transceivers [3].

In the last decade, many innovations were proposed to improve the performance of CT ΔΣ modulators and to address their main weaknesses: stability, sensitivity to jitter, feedback Digital to Analog Converter (DAC) mismatch and Inter-Symbol Interference (ISI).

Single-loop CT ΔΣ modulators have a simple architecture, but the noise-shaping order is limited by the stability of the loop. MASH architectures, which cascade lower-order single-loop ΔΣ modulators, have been developed since 1986 [4]. They are characterized by an aggressive noise shaping with few stability problems. In addition, an interstage gain can be used in the MASH to further suppress quantization noise [5]. However, they suffer from the mismatch between the analog loop filter and the digital noise-cancellation filters. Thus, the quantization noise leakage induced by this mismatch can significantly limit the final SNDR. Although classically it was mostly a parameter mismatch, in wideband implementations, all filter non-idealities matter. Another challenge is to accurately extract the quantization noise from previous stages as input to subsequent stages. Inaccurate quantization noise extraction also limits the overall resolution and may even lead to instability of subsequent steps. In [6], a CT sturdy MASH architecture was proposed to tackle both the stability problem and the mismatch problem with a very good resolution in a 50 MHz bandwidth. However, this architecture has stringent requirements in terms of Noise Transfer Function (NTF)/Signal Transfer Function (STF) design, which makes it sensitive to Process, Voltage and Temperature (PVT) variations. Another very interesting work presented in [7,8] uses a single-bit DAC to avoid the linearity problems of multi-bit DACs and tackles the jitter problem using a FIR DAC. This strategy leads to an incredible linearity with an SFDR > 100 dB but unfortunately it requires a high OverSampling Ratio (OSR), which puts high constraints on the clock generation and distribution. A third approach to improve the efficiency of CT ΔΣ modulators is using a noise-shaping SAR quantizer [9,10]. This reduces the number of analog active stages and results in incredible power consumption efficiency. However, this transfers the constraints on the high-resolution feedback DAC requiring the use of a separate high-voltage supply. Moreover, the delay and power-efficiency constraints on the digital part are high, making the use of this scheme interesting only in advanced nodes (28 nm and even 7 nm).

In this work, we propose a single-loop 5th-order 11-effective-bit 40 MHz CT ΔΣ ADC. The proposed architecture uses an OSR of only 8 to reduce the speed requirements on the clock generation, the loop and the decimation filter. The loop is stabilized by an innovative feedforward modulator scheme and a 5-bit flash quantizer with an integrated offset calibration. The ADC circuit integrates also a bandgap voltage reference that defines the ADC Full-Scale (FS) and a variable-rate decimation filter that reduces the modulator data rate from 640 MS/s to a user-defined output data rate from 20 MS/s to 80 MS/s. The conceptual architecture of the CT Delta Sigma ADC IC is shown in Figure 1.

The paper is organized as follows. Section 2 describes the system-level consideration that allows the implementation of a robust low-OSR CT ΔΣ modulator. Section 3 presents the circuit design. Section 4 provides the measurement results and Section 5 concludes the paper.

## 2. Delta Sigma ADC Architecture

### 2.1. Loop Architecture

The simplified architecture of the CT ΔΣ modulator is shown in Figure 2. An OSR of 8 is chosen to reduce the speed requirement of the loop filter and the multi-bit quantizer, which results in a low power consumption of the analog part. As the maximum input signal bandwidth is 40 MHz, the ΔΣ modulator is clocked at a fixed frequency of 640 MHz. The choice of the OSR value was made keeping in mind also the design of the digital decimation filter that may consume as much power as the ΔΣ modulator [11]. For example, the wideband ΔΣ modulator described in [7] achieves an outstanding power efficiency but the modulator sampling rate of 3.6 GS/s defers a serious challenge on the design of the digital decimation filter. Furthermore, lowering the switching-rate of the digital part reduces the crosstalk through the substrate.

To compensate for the lower noise-shaping performance at low OSR, a 5th-order NTF with a maximum gain (Qmax) of 12 dB and a 5-bit quantizer are combined to keep the quantization noise level well below the ADC noise floor, which is dominated by the thermal noise of the modulator front-end. A high Qmax value is also required to reduce the inherent tonal behavior of high-order ΔΣ loops working at low OSR, which results in a modulator with poor stability [12]. To address this issue, the signal feed-in paths $f_{i1}$, $f_{i4}$ and $f_{i5}$ are added. The feed-in coefficient $f_{i5}$ from the input of the modulator to the input of the quantizer has two purposes. First, it bypasses the loop filter that adds phase-shift to the input signal. The resulting “delay-free” path allows the quantizer to track the variation of the input signal faster. Second, together with the feed-in paths $f_{i1}$ and $f_{i4}$, they prevent most signal energy from leaking into the loop filter, which strongly reduces the voltage excursion at integrators outputs, as shown in Table 1. It results in a stable architecture that prevents from overload for full-scale input signals within the full bandwidth of the modulator, as illustrated in Figure 3a. The modulation index, defined as the ratio between the maximum stable amplitude (MSA) and the ADC full scale, is 0.85, which results in a MSA of −1.4 dBFS. Clocked at 640 MHz, the ΔΣ modulator achieves a signal-to-quantization noise ratio greater than 90 dBc over a 40 MHz bandwidth.

Table 2 displays the modulator coefficients, the coefficients were optimized as a compromise between the signal-to-quantization noise ratio and the implementation complexity (power consumption, feasibility, robustness). Although feed-in coefficients have the proven benefit of reducing the voltage swing inside the loop filter, it is rarely recalled that it is at the price of less alias rejection, as shown in Figure 3b. This is because feed-in coefficients flatten the STF out of the signal bandwidth.

In this design, the worst-case out-of-band component that may fall back within the signal bandwidth is attenuated by 42.3 dB. For comparison with what would be obtained using an anti- alias filter in front of the ADC IC, the inherent anti-alias filter provides the same attenuation at 600 MHz than a third-order Butterworth filter that has a maximum attenuation of 1 dB at the edge of the signal band.

Regarding the quantizer, while a 4-bit flash converter has been extensively used in previous wideband modulators [6,7,13], going to 5 bits reduces the maximum voltage swing inside the loop filter, which improves the compromise between power and distortion at low voltage supply. Having a 5-bit DAC also improves the stability and dynamic range tradeoff. As a matter of fact, as discussed in [14,15], increasing the quantizer number not just reduces the quantization noise floor but also reduces the quantizer gain variation, therefore enabling a better stability. Furthermore, the DAC sensitivity to jitter is reduced when the number of bits is increased, which allows the relaxation of the phase noise requirement on the system clock-source. To make the jitter noise contribution negligible, the rms value of the clock-source jitter must not exceed 1 ps. This value can be achieved with reasonable power consumption by an LC-PLL [11]. On the other hand, one of the main limitations of using a high-resolution quantizer is its input capacitance, which increases exponentially with the bit number and leads to high slew rate constraints on the last integrator. To address this problem, an offset calibration technique is used for the 5-bit quantizer allowing its input capacitance to keep significantly lower than the fifth-stage integrating capacitor. The calibration technique will be detailed in the next section.

Data Weighted Averaging (DWA) is chosen to correct mismatch-induced noise and distortion of the main DAC. As a high Qmax NTF is implemented, this technique is efficient even at low OSR [16]. An explicit loop delay of one half of the clock period is inserted in the feedback path to absorb the delay of the DWA and the intrinsic time response of the quantizer, which reduces the risk of having metastability-induced errors. This delay is compensated for with a feedback loop around the quantizer. To reduce capacitive loading at the virtual ground of the 5th integrator and relax speed requirement on its operational amplifier, the unit current cells of both DAC2 and DAC3 are scaled aggressively compared to those in DAC1. As a result, the DWA technique is also applied to these DACs to reduce their mismatch-induced spurs that are weakly attenuated by the loop filter at low OSR.

### 2.2. Decimation Filter

The decimation filter in a ΔΣ ADC ensures the transition from a low-resolution oversampled output to a high-resolution signal at the Nyquist rate while minimizing the aliasing of out-of-band noise into the useful band. In this work, the decimation filter also has an additional function. It needs to have a reconfigurable down-sampling ratio with 3 possible values 8, 16 and 32.

The chosen architecture for the decimator is shown in Figure 4. It operates as follows. After being converted from a thermometer code to a binary signal, the ΔΣ modulator output signal is filtered by a comb filter. This type of filter whose all coefficients are equal to one, is consequently very compact and has a low power consumption. However, it has the drawback of adding an in-band attenuation because its frequency response is $sinc^{k}\left(f\right)$ (sinc is the cardinal sine function and *k* is the filter order). The comb filter down-sampling ratio is reconfigurable, it can be adjusted to 4, 8 or 16. This point will be discussed in detail in the next paragraph. The decimated comb filter output signal is then applied to a half band (HB) filter. It is a class of symmetric Finite Impulse Response (FIR) filter whose cutoff frequency is $\frac{f_{s}}{4}$. By performing its inverse Fourier transform, one can note that one half of the filter coefficients are zeros, which divides its complexity by almost two. The HB output signal is then down-sampled by two to return to the Nyquist rate. Finally, the in-band attenuation introduced by the comb filter is corrected by a FIR symmetric equalizer.

Let us first focus on the comb filter. Its architecture is shown in Figure 5. As can be seen, the filter order was set to 6, one order higher than the modulator order. As in [17], the comb filter ensures the reconfigurability of the down-sampling ratio and therefore this has an impact on the number of bits of the wordlength. As a a matter of fact, the wordlength in comb filters is given by [18]:(1)$$WL_{comb}=n_{in}+k\times log_{2}\left(R\right)+1,$$
where $n_{in}$ is the input number of bits, which is 5 in this case, *R* is the down-sampling ratio, *k* the order. This leads to a $WL_{comb}$ of, respectively 18, 24 and 30 for the 3 down-sampling ratios of 4, 8 and 16. Therefore, $WL_{comb}$ was set to the maximum, i.e., 30 bits and to compensate for the down-sampling dependent wordlength, a reconfigurable bit-shift is added at the comb filter input. It is worth mentioning that the bit-shift could have been performed at the comb filter output instead; however, this approach increases the power consumption as unneeded togglings on the LSBs happen in the case of the down-sampling by 4 and 8.

For the HB filter, its order and the number of bits of its coefficients were determined to ensure the targeted resolution at the decimator output. The final block or the decimator is the equalizer, which corrects the in-band attenuation introduced by the comb filter, which depends slightly on its decimation order. The HB and equalizer coefficients were coded using the canonical signed digit (CSD) and were optimized using the approach presented in [19].

The performance was optimized for the fastest mode (80 MS/s) with a targeted ripple lower than ±0.05 dB. Nevertheless, as can be seen in Table 3, the ripple for the other modes was kept also low (<±0.2 dB). Table 3 also shows the SNDR simulation results at the decimator output. The estimated resolution of the decimation filter is, respectively 88, 92.1, and 92.2 dB for OSR 8, 16 and 32. These values are around 10 dB lower than the modulator resolution and therefore the degradation between the modulator output and the ADC output is lower than 0.5 dB.

## 3. Circuit Design

### 3.1. Loop Filter

Figure 6 shows the designed schematic of the 5-bit 5th order CT ΔΣ modulator. RC-integrators are preferred to Gm-C integrators for their higher linearity performance and robustness over PVT variations. The coefficient $f_{i5}$ is realized as the ratio of the feed-in capacitance $C_{f5}$ to the feedback capacitance of the last integrator, which avoids using an additional summing amplifier. The targeted Total Harmonic Distortions (THD) of 12 bits imposes a stringent linearity requirement on the modulator front-end because harmonic components introduced by the first integrator or the main DAC (DAC1), translate into harmonic degradation for the ADC as a whole. The noise floor of the modulator is dominated by the thermal noise and the flicker noise of the front-end that is composed of the input resistors, the opamp of the main integrator and the unitary current cells of the main DAC. The noise Power Spectral Density (PSD) of the front-end referred to the input of the modulator is computed by making the following assumptions and its derivation is detailed in Appendix A:The inverting/non-inverting input of the opamp are virtual groundsAll the noise source are uncorrelatedThe main contributor to DAC noise is the bottom NMOS transistor. The noise contribution of the NMOS cascode transistor and the switching transistors can be neglected.The main contributor to the opamp noise is the input transistors pair

This yields in the following expression: (2)$$\begin{array}{c} PSD\left(f\right)≃\underset{\mathrm{Resistor}}{\underset{︸}{8KTR_{1}}}+\underset{\mathrm{DAC}1}{\underset{︸}{16KTR_{1}\gamma \frac{Vref}{V_{gt-dac}}+\frac{KFI_{LSB}R_{1}^{2}}{fC_{ox}L_{u}^{2}}\alpha }}+\underset{\mathrm{Opamp}1}{\underset{︸}{\frac{8KT\gamma }{g_{m-opamp1}}+\frac{2KFI_{1}}{fC_{ox}L_{1}^{2}g_{m-opamp1}^{2}}}}. \end{array}$$The equation could be also organized by noise type as follows: (3)$$\begin{array}{c} PSD\left(f\right)≃\underset{\mathrm{Thermal}}{\underset{︸}{8KT\left(R_{1}+2R_{1}\gamma \frac{Vref}{V_{gt-DAC}}+\frac{\gamma }{g_{m-opamp1}}\right)}}+\underset{\mathrm{Flicker}}{\underset{︸}{\frac{KF}{fC_{ox}}\left(\frac{2I_{1}}{L_{1}^{2}g_{m-opamp1}^{2}}+\frac{I_{LSB}R_{1}^{2}}{L_{u}^{2}}\alpha \right)}} \end{array}$$
where *K* is the Boltzmann constant, *T* the absolute temperature, $R_{1}$ the input resistance of the first integrator, γ the noise factor (=2/3 for long channel devices), $g_{m-opamp1}$ the transconductance of the input stage of the first operational amplifier, $I_{1}$ the biasing current of input transistors of opamp1, $C_{ox}$ the gate-oxide capacitance per unit area, KF the flicker noise constant, α is a constant dependent on the technology parameters of $L_{1}$ the channel length of the input transistors of opamp1, Vref voltage reference of the quantizer, $L_{u}$ the channel length of DAC1 cell current source, $I_{LSB}$ DAC1 LSB current and $V_{gt-DAC}$ overdrive voltage of the unit current cells in DAC1.

Equation (Equation 3) shows that the flicker noise is dominated by the input stage of the opamp while the input resistors and the main DAC contribute the most to the thermal noise floor. For a given transconductance to current ratio, the flicker noise can be reduced by increasing the area of the devices up to a certain point (limited by parasitic capacitances). Regarding thermal noise, (Equation 3) shows that a fundamental limit exists that is set by the resistor value and a scaling factor whose value depends on the ratio of the reference voltage of the quantizer to the overdrive voltage of the current cells in the main DAC. In this design, the PSD of the flicker noise is set to 30 nV/$\sqrt{\mathrm{Hz}}$ at 10 kHz while the overdrive voltage of the current cells is set to 0.25 of the supply voltage to optimize noise and matching performance. Figure 7 shows the contribution of each component to the noise PSD and the rms noise (integrated from 10 kHz to 40 MHz) contribution of each component in the front-end. The noise PSD predicted by (Equation 3) fits very well with the simulation, which makes it useful to budget the noise in a given process.

Figure 8a shows the simplified schematic of the ΔΣ modulator front-end. The finite Gain Bandwidth Product (GBW) induced voltage swing at the virtual ground of the first opamp (opamp1) is the main source of distortion in the front-end. This residual voltage, which is signal-dependent, modulates the input transconductance (gm1) of opamp1 and the finite output impedance of DAC1, which creates harmonic distortion. In many prototypes, the value of the finite GBW of the first opamp is chosen equal to the sampling frequency and the original shape of the NTF is restored by coefficient tuning [20]. This design strategy guarantees high power efficiency at the cost of linearity performance because lowering the GBW increases the voltage swing at the virtual ground of opamp1. Moreover, even if the finite-GBW phase-shift through the loop filter is rigorously compensated for by tuning components values [20], the robustness of the modulator stability against PVT variations cannot be guaranteed. In this work, we decided to adopt a more robust approach at the cost of higher power consumption. The first integrator embeds a 4-stage amplifier that achieves a GBW and a phase margin of 4 GHz and 70 degrees, respectively, which ensures low-distortion and low phase-shift against PVT variations. The last stage (gm4) uses minimum length devices to increase the phase margin while consuming low power. The main opamp consumes 20 mA and its third-order harmonic amplitude at the modulator output is 100 dB below the ADC Full Scale. This guarantees a THD dominated only by the current mismatch of DAC1. A scaled version of opamp1 is used in the 5th integrator while 3-stage opamps are implemented in the other integrators to save area and power. To compensate for the shifting of the RC time-constant that can reach ±35% in this process, the feedback capacitors of each integrator are tuned to the ideal value with an accuracy of ±2% [21]. To achieve such good accuracy, the smallest needed capacitance needs to be as small as 5 fF. It was built by fitting in series 8 capacitors of 40 fF. To ensure a good matching between RC passives of the tuning circuit and those of the integrators each integrator has a dedicated on-chip auto-tuning circuit.

### 3.2. DACs and Quantizer

The architecture of one of the 31-unit current cells used in DAC1 is shown in Figure 8a. It consists of a regulated cascode current source whose bottom transistor is sized to meet the matching requirement for a THD > 12 bits. The boosting amplifier is optimized to maximize the output impedance of the DAC over a wide frequency range. An impedance value higher than 400 MΩ is obtained up to the band edge (40 MHz), which makes the distortion mechanism due to the finite GBW of opamp1 negligible within the signal bandwidth. A careful design of the clock distribution circuit and flip-flops that drive DAC switches has led to an additive jitter value of only 137 fs rms. This value leaves enough margin for the jitter of the clock-source. As said earlier, the loop delay is compensated with two DACs that are merged with the intrinsic feedback-DAC connected to the virtual ground of the 5th integrators [11]. Because mismatch-induced noise and distortion of both DAC2 and DAC3 are noise shaped by the loop filter, their unit current cells are scaled aggressively compared to those in DAC1. This reduces the capacitive loading at the virtual node of opamp5. As also specified earlier, a DWA algorithm is used to address DAC mismatch. The available delay for the whole feedback loop including the quantizer, the DWA and the DAC drivers is $T_{s}$/2. The use of a low OSR in this design relaxes this delay constraint as $T_{s}$/2 is 781 ps. The DWA, therefore, was designed using standard cells. Its delay is 172 ps for a power consumption of 0.82 mW.

The flash quantizer consists of 31 comparators, which convert the output of the loop filter into a thermometric code plus two comparators that detect positive or negative signal excursions, which might saturate the quantizer. The schematic of one comparator slice is detailed in Figure 8b. A voltage buffer (not shown) provides a copy of an on-chip bandgap reference to a resistive ladder, which generates the threshold voltages Vref[k]. Compared to the architecture proposed in [11], our pre-amplifier uses a non-switching load, which strongly minimizes the “kickback” effect on the loop filter output and the reference ladder. The devices inside the comparator are sized small (about 0.14 μm^2^) and the mismatch-induced offset that is randomly distributed from −80 mV to +80 mV (±3σ) before calibration is compensated at the same time for each comparator during start-up. The offset value is estimated by a feedback loop, which consists of a 5-bit DAC and a counter/integrator as shown in Figure 8b.

For each DAC code, the current ($I_{DAC}$) is incremented and injected in differential to the input pair NMOS transistors of the comparator pre-amplifier. The output of the comparator is integrated over several fractions of the clock period to reduce the influence of noise. When the mean value of the comparator output is equal to or greater than zero, then the counter stops and leaves the output current of the DAC at a value that compensates for the offset voltage.

After calibration, the residual offset is bounded within one DAC’s LSB.

Compared to [11], each comparator is calibrated around its trip point rather than around its input common mode, which has the benefit of correcting both static and dynamic offsets. The input capacitance and the power consumption of the quantizer are 50 fF and 5.5 mW, respectively. In normal mode (calibration off), the on-state resistance of the switch that connects the loop filter to the quantizer, together with the input capacitance of the quantizer create a pole whose value must be set well beyond the sampling frequency of the modulator to preserve its stability.

### 3.3. Decimation Filter

The decimator was synthesized using the 1 V GPLVT library of the STMicroelectronic process. Post-layout back-annotated simulations were carried out and match perfectly the RTL simulations.

The decimator gate count and the area are reported in Table 4. As can be seen, the decimation filter die area is only 0.122 mm^2^.

Table 5 shows the power consumption of the decimation filter for the 3 scenarios. As excepted, the OSR = 8 has the highest consumption as in this case, the equalizer and the HB filter run twice as fast with respect to the OSR = 16 case and four times faster with respect to the OSR = 32 case. To estimate the overhead induced by reconfiguration, a design optimized only for the OSR = 8 case was also built. The obtained die area is 0.11 mm^2^ and the power consumption is 11.37 mW, which correspond, respectively, to an overhead of 10% of die area and 5% of power consumption.

## 4. Prototype and Measurement Results

### 4.1. General

A prototype was fabricated in 65 nm 1P7M CMOS process. The chip micrograph is shown in Figure 9 (left). It is encapsulated in a 100-pin Ceramic Quad Flatpack (CQFP) package. The die area is 2.35 mm^2^ including the ΔΣ modulator, a bandgap reference, the programmable decimation filter, all tuning and calibration circuits and high-speed Current-Mode Logic (CML) buffers. These latter were added in the IC to capture the high-speed modulator 5-bit digital output. They were preferred to classical CMOS buffers because they induce a significantly lower ringing on the power supply line. For the 16-bit decimator output, the I/O ring digital cells are fast enough to drive the analyzer probe for an output of 80 MS/s and therefore no extra buffers were needed for this signal. Analog and digital blocks use 1.2 V and 1 V of voltage supply, respectively. Digital blocks were isolated in a deep N-well to reduce coupling with the analog blocks through the substrate. The sensitive analog blocks such as the bandgap and the master bias circuit were placed at the top left part of the chip, far from the components that have a high switching activity. Tiling is avoided above analog blocks that require good matching.

The test setup used to evaluate the dynamic performance of the prototype ADC is shown in Figure 9 (right). An analog signal source (R&S AFQ100A) drives a sine wave tone into a passive bandpass filter (K&L D5BT-6/12) of which the central frequency can be swept over the whole signal bandwidth of the ADC. An RF transformer (Mini-Circuits ADT1-6T+) converts this spectrally purified tone into a differential signal that serves as the input to the prototype ADC. The clock signal is provided by a wideband generator (R&S SMA 100A), which can generate low-jitter sine waveform (50 fs rms in the bandwidth of interest). The modulator and decimator outputs by the prototype ADC are acquired by of a high-speed logic analysis system (Agilent 16901) that has a memory depth of 4 MB. Stored data are then post-processed with MATLAB on a PC. The RC time constant of the integrators and the offset of the comparators are auto-calibrated once after the test setup power up. No calibration is redone during the measurement procedure.

### 4.2. Measurement Results

Unfortunately, a major error has limited our measurements. When DWA was turned on, the noise floor and the even harmonic distortions increased. To understand this problem, extensive research into the literature and investigations were carried out. It has been found that the main reason for the observed problem is the superposition of ISI in the DACs with the use of the DWA algorithm. Due to the mismatch between the switching components of the DAC positive and negative terminals, the current pulse delivered by each unitary cell of a current-steering DAC has unequal rise and fall time. This imbalance creates an error whose magnitude becomes higher when the number of transitions increases. The DWA operation consists of increasing the number of transitions to average the DAC current mismatch; however, this emphasizes the ISI error and, depending on the values of the DAC current mismatch and ISI, using a DWA can either improve or degrade the performance. Therefore, for the presented measurement results, the DWA was turned off.

As shown in Figure 10a, the ADC achieves a dynamic range of 71.4 dB. Figure 10b shows the PSD of the Delta Sigma modulator output for a −2.5 dBFS input. The measured peak Signal-to-Noise and Distortion Ratio (SNDR) and peak THD are 68.6 dB and 76.5 dBc, respectively, which results in 11-bit Effective Number Of Bits (ENOB) in a 40 MHz bandwidth. The inter-modulation products are measured by injecting at the input of the modulator two sinusoidal signals of equal amplitude (−8.5 dBFS) and whose frequencies are 10 and 11 MHz, respectively. The worst-case IM2 and IM3 are, respectively, 79.2 and 79 dBc. The intrinsic alias rejection was measured with a sinusoidal input signal from 600 MHz to 640 MHz, the first band that could alias on the useful band. The measurements show an attenuation higher than 40 dB on all the range as predicted by behavioral simulations, confirming that the modulator loop filter provides alias rejection equivalent to a 3rd-order Butterworth filter.

Figure 11 shows the PSDs at the modulator and decimator outputs for the same measurement carried out with an OSR of 16. The input signal is a 1 MHz input sine at −2.5 dBFS. As can be seen, the two spectrums completely match in the [0–20 MHz] bandwidth proving that the decimation filter works as expected and the SNDR degradation is lower 0.1 dB. Similar results were obtained for the other OSR configurations. Figure 12 shows the frequency response of the decimator in an OSR = 8 configuration. To achieve this measurement, the gain was measured at both the modulator and decimator outputs. The plotted curve is the ratio of the 2 curves to just show the decimator gain. As can be noticed, the ripple is ±0.09 dB, which is slightly higher than the simulated ripple ±0.04 dB.

## 5. Conclusions

Table 6 compares this work to similar works from the literature. The proposed ADC Figure Of Merit (FOM)s are not as good as the best works in the literature due mainly to the aforementioned DWA problem. Besides this aspect, the proposed ADC performance could be improved by decreasing its power consumption. To achieve this goal, several ideas could be investigated. First, the power consumption of the loop filter could be scaled using Gm-C integrators after the main opamp-RC integrator. As the proposed architecture benefits of very low swing inside the loop filter, using gm-C integrators would greatly improve the power efficiency without significantly affecting the overall linearity. The second block that contributes mostly to the overall power consumption of the modulator is the quantizer. A highly digital quantizer as in [22] could improve the power efficiency compared to a regular FLASH ADC. Furthermore, the reduced number of comparators and the digital implementation of the excess-loop delay compensation could reduce the die area significantly. As an extra DAC is avoided, the capacitance at the virtual ground of the last opamp is reduced, which translates to further reduction of power.

Nevertheless, this work has several interesting features. First it has the specificity, similarly to [11], of being a midway between a prototype and a product since it integrates a very low ripple (<0.1 dB) digital decimation filter and a bandgap. Second, the integrator time constants and quantizer offset calibrations are done on-chip with very good efficiency and precision. The proposed calibrations reduce the time-constant variation from 30% to 2% and the 3σ comparator offset from ±80 mV to less than 5 mV. Finally, the proposed ADC also integrates an innovative 5th-order loop structure that allows the use of an OSR as low as 8. This greatly relaxes the constraints on the clock generation and distribution in the IC.

## Figures and Tables

**Figure 1 sensors-23-00036-f001:**
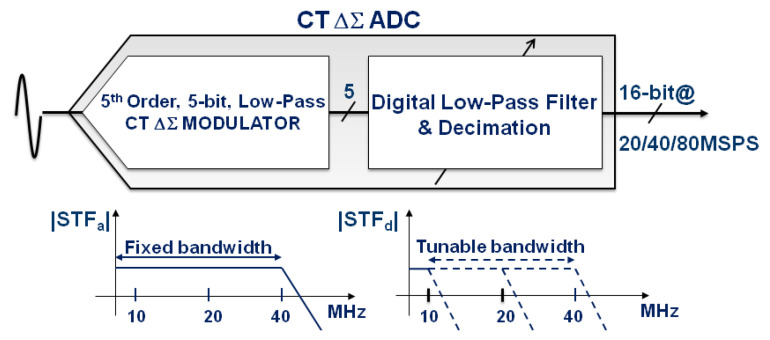
Architecture of the proposed variable-rate CT ΔΣ ADC IC.

**Figure 2 sensors-23-00036-f002:**
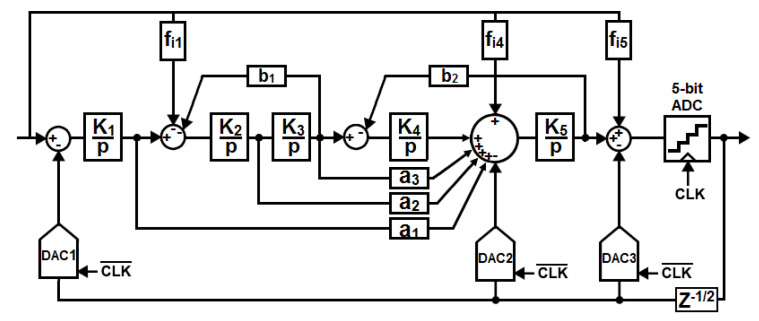
Architecture of the proposed ΔΣ modulator. (p is the Laplace variable).

**Figure 3 sensors-23-00036-f003:**
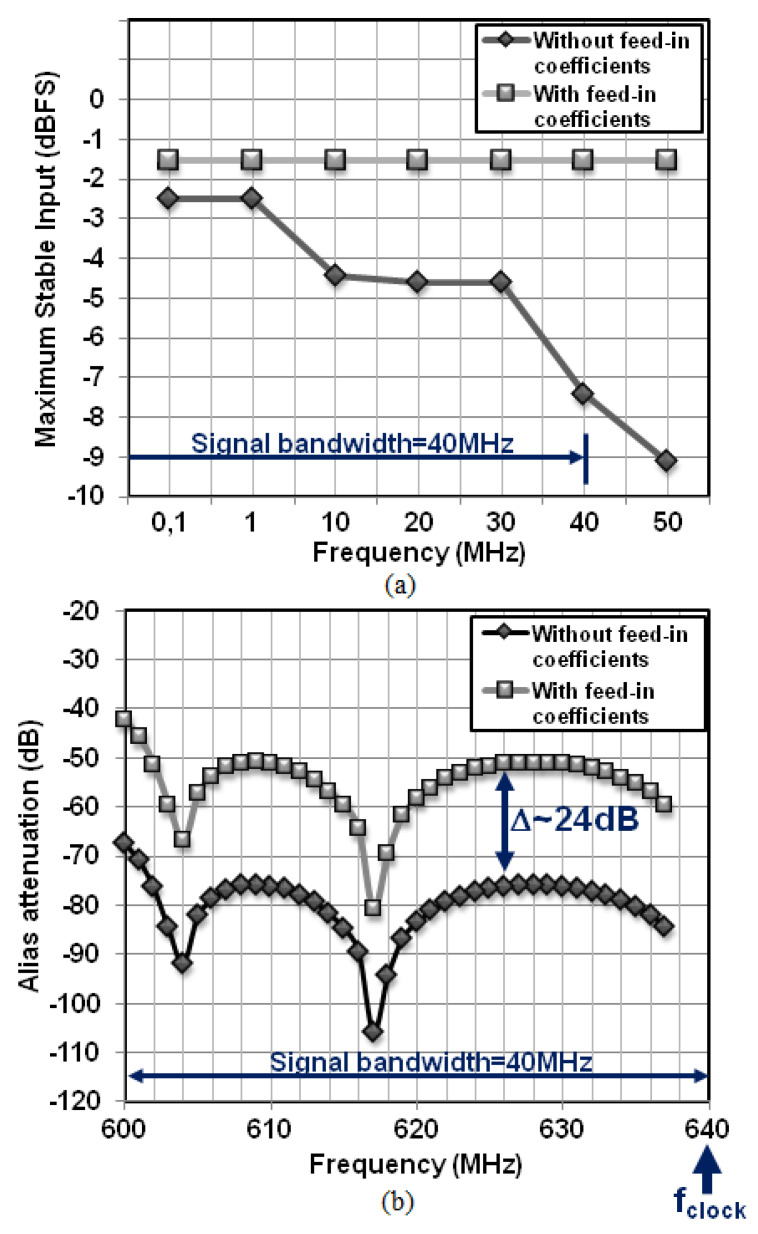
Feed-in coefficients effect on (**a**) the maximum stable input amplitude, (**b**) the inherent anti-alias filter transfer function. (The attenuation curves are obtained with transient simulations followed by a frequency analysis).

**Figure 4 sensors-23-00036-f004:**
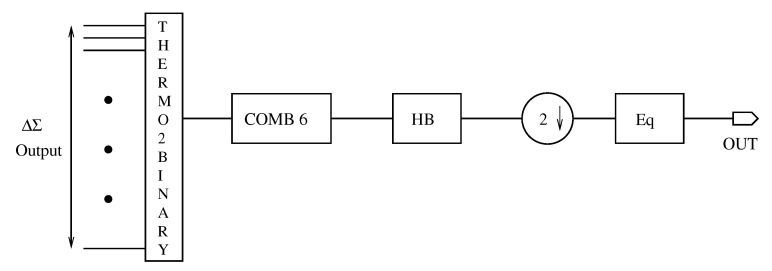
Block diagram of the decimation filter.

**Figure 5 sensors-23-00036-f005:**
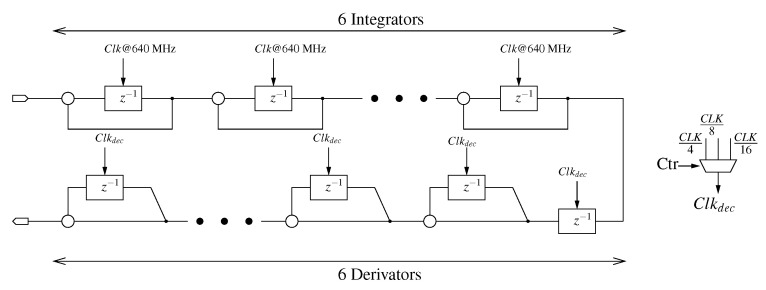
Comb filter.

**Figure 6 sensors-23-00036-f006:**
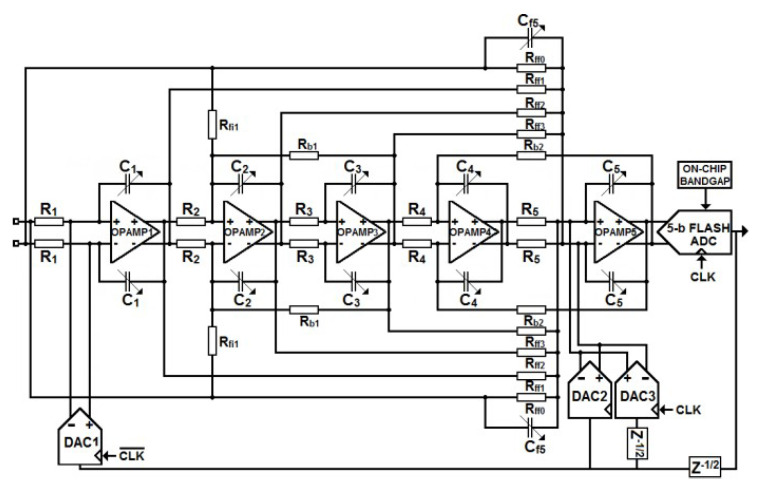
Circuit-level architecture of the CT ΔΣ modulator.

**Figure 7 sensors-23-00036-f007:**
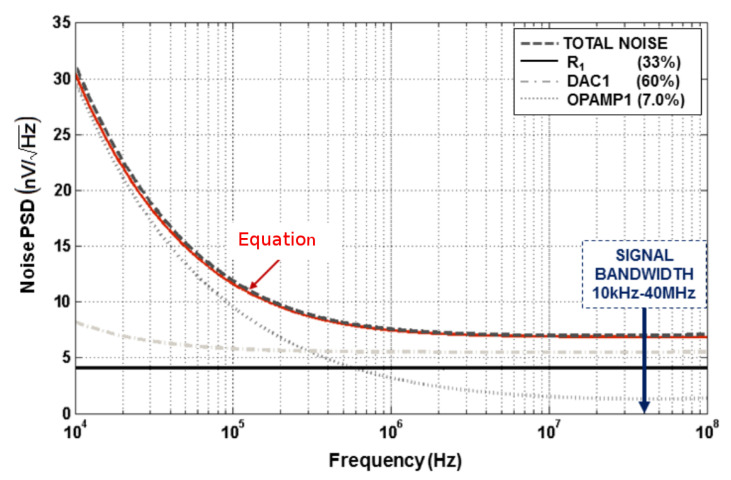
Breakdown of the noise contribution in the modulator front-end.

**Figure 8 sensors-23-00036-f008:**
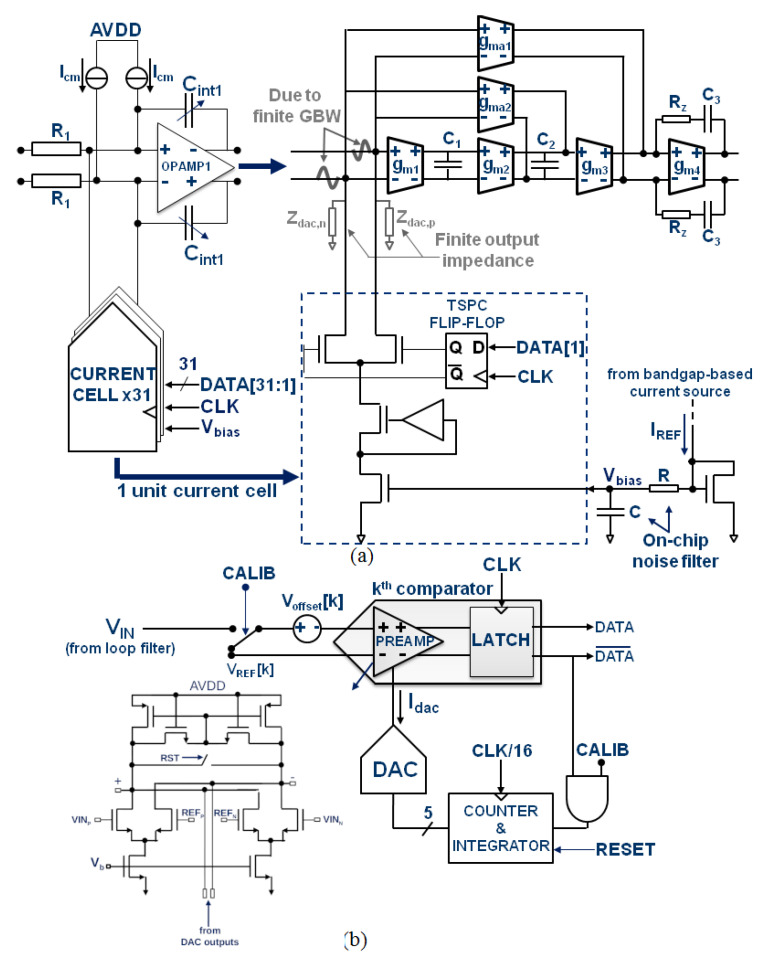
Schematic of (**a**) the ΔΣ modulator front-end, (**b**) one comparator cell.

**Figure 9 sensors-23-00036-f009:**
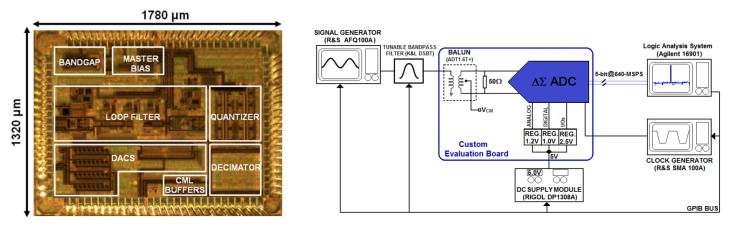
(**Left**) Chip micrograph. (**Right**) Measurement setup.

**Figure 10 sensors-23-00036-f010:**
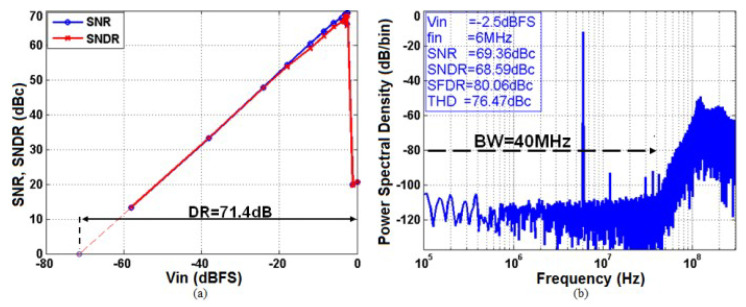
(**a**) PSD of the ΔΣ modulator output (65536 FFT points). (**b**) Measured SNR and SNDR as a function of the input signal level (finput = 6 MHz).

**Figure 11 sensors-23-00036-f011:**
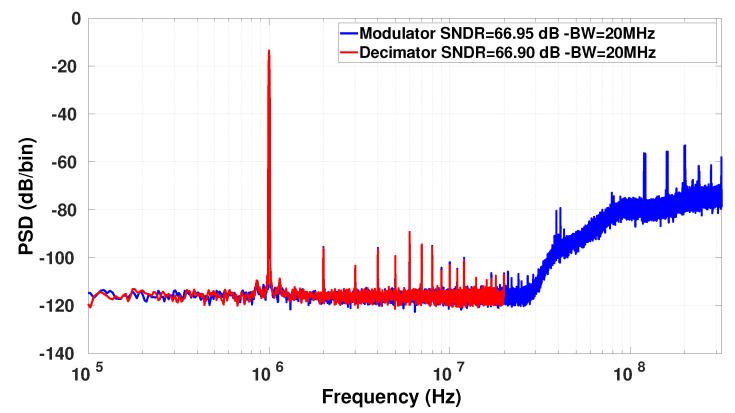
Spectrums of the modulator and decimator outputs for an OSR of 16.

**Figure 12 sensors-23-00036-f012:**
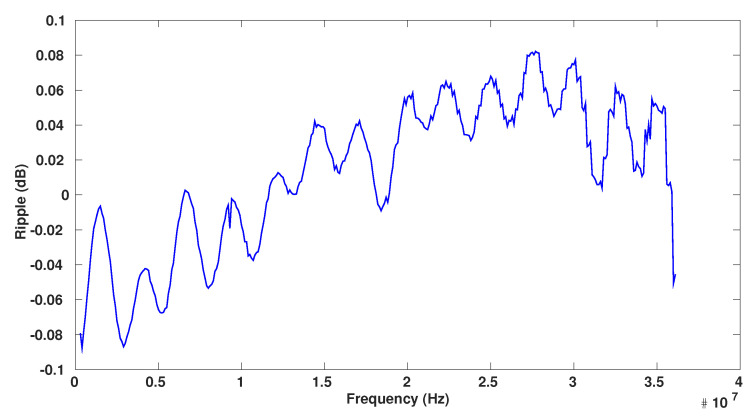
Frequency response of the decimation filter for an OSR of 8.

**Table 1 sensors-23-00036-t001:** Maximum output swing of the modulator $f_{in}=$10 MHz $A_{in}$= −4.4 dBFS Full-scale Quantizer FS = 800 mV.

Configuration	Without Feed-In	With Feed-In
Integrator 1	±261 mV	±235 mV
Integrator 2	±247 mV	±67 mV
Integrator 3	±313 mV	±52 mV
Integrator 4	±450 mV	±48 mV
Integrator 5	±766 mV	±421 mV

**Table 2 sensors-23-00036-t002:** Modulator coefficients.

Order	k1	k2	k3	k4	k5	a1
5	1.44	0.69	0.44	0.22	8.90	5.93
**a2**	**a3**	**fi1**	**fi4**	**fi5**	**b1**	**b2**
5.93	4.45	0.17	6.67	2	0.46	0.022

**Table 3 sensors-23-00036-t003:** Decimation filter parameters. *The ripple is simulated for $f_{in}<0.9\;Bw$*.

OSR	${SNR}_{dec}$	${SNDR}_{mean}$	Ripple
		ADC Decimated	
8	88.01 dB	77.14 dB	±0.04 dB
16	92.1 dB	80.32 dB	±0.13 dB
32	92.2 dB	83.22 dB	±0.18 dB

**Table 4 sensors-23-00036-t004:** Decimator gate count and area.

	Comb^6^	HB	Equalizer	Overall
nb of gates	1449	1380	5349	8193
area (mm^2^)	0.025	0.02	0.077	0.122

**Table 5 sensors-23-00036-t005:** Decimator power consumption split.

	Leakage (mW)	Dynamic (mW)	Total (mW)
OSR = 32 Comb	0.175	4.871	5.046
HB	0.115	0.354	0.469
EQ	0.375	1.134	1.506
Dec	0.665	6.56	7.02
OSR = 16 Comb	0.175	4.825	5.001
HB	0.115	0.719	0.834
EQ	0.375	2.381	2.755
Dec	0.665	8.168	8.833
OSR = 8 Comb	0.175	4.820	5.001
HB	0.115	1.450	1.519
EQ	0.375	4.789	5.163
Dec	0.665	11.248	11.913

**Table 6 sensors-23-00036-t006:** Performance table and comparison to the state of the art. $FOM_{W}=\frac{P}{2\cdot Bw\cdot 2^{\frac{SNDR-1.76}{6.02}}}$$FOM_{S}=DR+10\log\left(\frac{Bw}{P}\right)$.

	Xing-20 [10]	Lo-19 [9]	He-18 [13]	Wu-16 [23]	Dong-14 [24]	Mit.-06 [11]	This Work
**Architecture**	**2nd Order** **Loop with** **4-Bit SAR**	**1st Order** **Loop with** **7-Bit SAR**	**4th Order** **Loop with** **ISI Calib**	**6th Order** **Noise** **Coupling**	**3-1 Sturdy** **MASH**	**3rd Order** **Loop with** **4-Bit Flash**	**5th Order** **Loop with** **5-Bit Flash**
Process (nm)	28	7	28	65	28	130	65
$f_{s}$ (MHz)	1560	400	2000	900	3200	640	640
BW (MHz)	50	25	50	45	45	20	40
OSR	15	8	20	10	35	16	8
DR (dB)	80.6	79.4	82.8	82.5	90	80	71.4
SNDR (dB)	74.4	74.0	79.8	75.3	72.6	74	68.6
P. Mod. (mW)	10.4	3.8	64.3	24.7	235	20	82.3
P. Dec. (mW)	-	-	-	-	-	20	12.3
FOM_*W*_ (fj/st.)	24.2	18.6	80.5	57.7	748	120	503
FOM_*S*_ (dB)	177.1	177.6	171.7	167.9	172.9	170	158.1

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
