# Peer review of "A 40 MHz 11-Bit ENOB Delta Sigma ADC for Communication and Acquisition Systems"

_sensors, 2022, doi:10.3390/s23010036_

Round 1
Reviewer 1 Report
The overall organization of this paper is not clear. However, I have some concerns about this manuscript.
1. The introduction is brief; there is room for improvement.
2. Where is Idac go to pre-AMP in Fig8(b)?
3. Please provide measured result for SNR and SNDR as a function of the input signal level at highest frequency.
4. Please explain more clearly how to achieve low-power ADCs. The proposed circuit requires high-speed OP.
Author Response
The answer is to review as well as the revised version are in attachment

Reviewer 2 Report
General Comments:
The paper deals with delta-sigma modulators and presents a design for an integrated delta-sigma ADC with a 5th-order continuous-time delta-sigma modulator. The proposed SDM-ADC works at a low OSR of 8, which eases some requirements on the circuit design. The proposed design could be useful for modern DSP and communication systems. However, clarity of presentation is a big issue in this paper. As insufficient clarity can obscure the contribution of the paper and limit its reference value, hence, major amendments are required to improve the presentation.
Specific Comments:
1. Section 2: Almost all ideas related to the proposed design in Figure 2 are presented without sufficient explanation, referencing, or reasoning.
(a) It is unclear how an OSR of 8 is considered as sufficient for high date-rate communication. Is the input signal assumed to be slowly-varying or highly-correlated?
(b) It is unclear how a 5th-order filter can compensate for the loss of noise-shaping performance. Explicit noise and signal transfer functions are necessary, along with the filter transfer function and its coefficients.
(c) It is known that higher-order loop filters cause stability problems in DSMs. Please analyze the stability of the proposed design, and clarify why a 5th-order is chosen.
(d) Is the 5-bit quantization in Figure 2 related to the 5th-order filtering?
(e) Is there a fractional delay (i.e., z^{-1/2}) in Figure 2?
(f) In Figure 2, it is unclear what “p” stands for.
(g) Please explain how the attenuation curves in Figure 3 are obtained.
(h) Please explain the meaning of the modulation index (which was chosen as 0.85).
(i) It is unclear how a 5-bit DAC can improve the stability and dynamic range. Please clarify or put a Reference. This comment applies to all un-explained statements in this paper.
2. Section 3: Please justify the PSD in Equation (2).
Author Response

(The authors gave the same response as above.)

Round 2
Reviewer 2 Report
The Authors have carefully addressed all of the Reviewer’s concerns. The revised version is useful and suitable for publication in Sensors.
Note: It would be better to add the derivation of the PSD as an Appendix.
Author Response
Thank you for the feedback. The PSD derivation has been added in appendix as recommended by the reviewer.